# Association of Patients’ Epidemiological Characteristics and Comorbidities with Severity and Related Mortality Risk of SARS-CoV-2 Infection: Results of an Umbrella Systematic Review and Meta-Analysis

**DOI:** 10.3390/biomedicines10102437

**Published:** 2022-09-29

**Authors:** Eduardo Reyna-Villasmil, Maria Giulia Caponcello, Natalia Maldonado, Paula Olivares, Natascia Caroccia, Cecilia Bonazzetti, Beatrice Tazza, Elena Carrara, Maddalena Giannella, Evelina Tacconelli, Jesús Rodríguez-Baño, Zaira R. Palacios-Baena

**Affiliations:** 1Unit of Infectious Diseases and Clinical Microbiology, University Hospital Virgen Macarena, Institute of Biomedicine of Seville (IBIS)/CSIC, 41008 Seville, Spain; 2Department of Medical and Surgical Sciences, University of Bologna, 40126 Bologna, Italy; 3Infectious Diseases Unit, IRCCS Azienda Ospedaliero-Universitaria di Bologna, 40126 Bologna, Italy; 4Infectious Diseases Division, Department of Diagnostics and Public Health, University of Verona, 37129 Verona, Italy; 5Department of Medicine, University of Seville, 41008 Seville, Spain; 6Centro de Investigación Biomédica en Red en Enfermedades Infecciosas (CIBERINFEC), Instituto de Salud Carlos III, 28029 Madrid, Spain

**Keywords:** COVID-19, SARS-CoV-2, meta-analysis, mortality, severe disease, predictors, comorbidities

## Abstract

The objective of this study was to assess the association between patients’ epidemiological characteristics and comorbidities with SARS-CoV-2 infection severity and related mortality risk. An umbrella systematic review, including a meta-analysis examining the association between patients’ underlying conditions and severity (defined as need for hospitalization) and mortality of COVID-19, was performed. Studies were included if they reported pooled risk estimates of at least three underlying determinants for hospitalization, critical disease (ICU admission, mechanical ventilation), and hospital mortality in patients diagnosed with SARS-CoV-2 infection. Evidence was summarized as pooled odds ratios (pOR) for disease outcomes with 95% confidence intervals (95% CI). Sixteen systematic reviews investigating the possible associations of comorbidities with severity or death from COVID-19 disease were included. Hospitalization was associated with age > 60 years (pOR 3.50; 95% CI 2.97–4.36), smoking habit (pOR 3.50; 95% CI 2.97–4.36), and chronic pulmonary disease (pOR 2.94; 95% CI 2.14–4.04). Chronic pulmonary disease (pOR 2.82; 95% CI 1.92–4.14), cerebrovascular disease (pOR 2.74; 95% CI 1.59–4.74), and cardiovascular disease (pOR 2.44; 95% CI 1.97–3.01) were likely to be associated with increased risk of critical COVID-19. The highest risk of mortality was associated with cardiovascular disease (pOR 3.59; 95% CI 2.83–4.56), cerebrovascular disease (pOR 3.11; 95% CI 2.35–4.11), and chronic renal disease (pOR 3.02; 95% CI 2.61–3.49). In conclusion, this umbrella systematic review provides a comprehensive summary of meta-analyses examining the impact of patients’ characteristics on COVID-19 outcomes. Elderly patients and those cardiovascular, cerebrovascular, and chronic renal disease should be prioritized for pre-exposure and post-exposure prophylaxis and early treatment.

## 1. Introduction

By the end of April 2022, the COVID-19 pandemic, caused by the new coronavirus SARS-CoV-2, had caused more than 500 million cases and more than 6 million deaths worldwide [1]. SARS-CoV-2 frequently causes asymptomatic or mild infection; however, some patients progress to a severe disease, which is associated with high mortality [2]. Identifying the patients’ conditions associated with the development of severe forms of COVID-19 and mortality is helpful because it allows identifying the patients who would benefit most from specific preventive interventions, including enhanced transmission-protective measures, being prioritized in vaccination campaigns, and, more recently, receiving antivirals or monoclonal antibodies, which may avoid the progression from mild to severe disease.

Several patient conditions, including age, gender, and several chronic underlying comorbidities, have been associated with worse outcomes [2]. However, the generalizability of the estimates for the relative impact of each of these conditions in the different studies performed may be hampered because the different studies may be affected by selection and information biases and lack of statistical power. As a result, the estimations of the risk associated with underlying conditions of the patients for the development of severe COVID-19 or mortality are frequently heterogeneous, if not contradictory.

The aim of this study was to conduct an umbrella systematic review and meta-analysis in order to assess the association between patients’ epidemiological characteristics and comorbidities with severity and related mortality risk of SARS-CoV-2 infection.

## 2. Methods

### 2.1. Design, Data Sources, and Search Strategy

An umbrella literature review was conducted according to the Joanna Briggs Institute recommendations [3] and reported according to Preferred Reporting Items for Systematic Reviews and Meta-Analyses (PRISMA) [4]. The study protocol was registered in PROSPERO (CRD42021267368). Patients were not involved in the design, conduct, interpretation, and writing up of the results of this study.

For this umbrella review, the PICO question was defined as follows. The patients were outpatients and inpatients diagnosed with SARS-CoV-2 infection; the exposures were the epidemiological characteristics and chronic underlying conditions of the patients; the comparator was the absence of exposure to these characteristics and conditions; and the outcomes considered were hospital admission, severe/critical COVID-19, and in-hospital mortality. Published systematic reviews and meta-analyses on the association of patients’ epidemiological characteristics and comorbidities with hospitalization due to COVID-19, development of severe or critical COVID-19 defined as mechanical ventilation and need of intensive care unit (ICU) admission, and death were considered only in admitted patients with a first episode of infection.

The literature search was conducted in PubMed, MEDLINE, EMBASE, Web of Science, Scopus, Cochrane Library databases, and the JBI database of Systematic Reviews and Implementation Reports, with no language restrictions. The initial search was conducted on 1 August 2021 and updated on 30 September 2021. The full search strategy used is shown in Appendix A.

### 2.2. Inclusion and Exclusion Criteria

Articles were eligible if they were published between December 2019 and August 2021 and if they included a meta-analysis of the association of patients’ epidemiological characteristics and comorbidities with the severity or mortality from COVID-19. Studies had to meet the following criteria: (a) they were conducted on patients diagnosed with SARS-CoV-2 infection by PCR or antigen test in nasopharyngeal or respiratory tract samples; (b) they included the evaluation of at least three epidemiological characteristics and comorbidities in order to be able to assess the confounding effect of one condition on others; (c) they provided quantitative data of patients with and without the conditions and their outcomes; (d) they provided a pooled estimation of the association of the conditions and the outcomes. Studies in which severity or mortality was not the primary outcome, narrative reviews, meta-analyses including fewer than 5 studies, and preprints were excluded. Systematic reviews reporting outcomes in vaccinated patients or in pregnant women and children (aged less than 18 years) were also excluded, as these groups may have specific outcome determinants.

### 2.3. Article Selection and Data Extraction

All the identified references were managed with a reference management program, and duplicates were removed. The titles and abstracts were screened, and the full texts of the selected articles were then reviewed for eligibility and data extraction by two investigators (ZRP-B and ER-V). A third coauthor (JR-B) resolved any disagreement that could not be resolved by consensus. The data extracted included: author, year of publication, number of participants, number and type of studies included, quality assessment instrument used, method of analysis, patients’ epidemiological characteristics and comorbidities and outcomes assessed, heterogeneity, and the estimated associations between all conditions and the outcomes. The AMSTAR 2 tool [5] was used to assess methodological quality and assign an overall rating for the reviews included Appendix A.

The patients’ epidemiological characteristics and comorbidities were grouped into categories. The definitions used in the included systematic review were reviewed for homogeneity. The outcomes considered were hospitalization due to COVID-19, development of severe/critical disease (i.e., need for ICU admission, high-flow oxygen or mechanical ventilation), and mortality. Data of the association of the conditions and outcomes were collected as rate ratios (RR), odds ratios (OR), and hazard ratios (HR), with 95% confidence intervals (CI).

### 2.4. Data Analysis

The characteristics and results of the included studies were synthesized and presented in tables and forest diagrams. Pooled OR (pOR) with 95% confidence intervals (CI) were calculated for conditions investigated in at least 3 meta-analyses using the DerSimonian and Laird random-effects method, which accounts for inter- and intra-study variance. For dichotomous variables, a summary of estimations was produced by using a logarithmic scale to maintain symmetry in the analysis. An estimate of publication bias was calculated with Egger’s regression test. The I^2^ statistic was used as an estimate of true variance in the summary estimate and was used as an estimate of the proportion of variance reflecting the true differences in effect size. The degree of overlap of primary studies included in the different meta-analyses was investigated by a citation matrix including the systematic reviews in columns and the primary studies included in rows; the degree of overlap was quantified using the corrected covered area (CCA). The overlap was categorized as very high (>15%), high (11–15%), moderate (6–10%), or light (0–5%) [6]. CCA is a validated method for quantifying the degree of overlap between two or more reviews.

## 3. Results

The initial search identified 411 potentially eligible studies. After discarding duplicates, 225 were screened, and finally, 16 systematic reviews and meta-analyses were included [7,8,9,10,11,12,13,14,15,16,17,18,19,20,21,22] (Figure 1). These systematic reviews included 568 primary studies, with a range of 7 and 77 per meta-analysis.

The characteristics of the selected studies are shown in Table 1. Overall, the risk estimates for the association of 12 underlying patient conditions with some of the outcomes in patients diagnosed with COVID-19 were available, including: age, sex, smoking status, obesity, hypertension, diabetes mellitus (DM), cardiac disease (CD, including arrhythmia or chronic heart failure), chronic pulmonary diseases (CPD), cancer (hematological cancer, solid cancer, any malignant tumor), cerebrovascular diseases (CVD, including stroke and transient ischemic attack), chronic kidney disease (CKD), and chronic liver disease (CLD). Seven systematic reviews provided information on the risk of hospitalization for 11 characteristics and comorbidities (all except CLD) [7,8,9,10,11,16,22]. Nine reported the risk of severe/critical illness from 10 determinants (all except age and hypertension) [10,11,12,13,14,15,16,17,18], and six reported risk estimates for the mortality ratio of 11 factors (all except age) [14,17,18,19,20,21]. Overall, four considered adjusted estimations for the individual conditions [11,14,18,19].

Thirteen of the sixteen systematic reviews and meta-analyses were rated as high quality [8,10,11,12,13,14,15,19,20,21,22], two were considered of moderate quality [7,16], and one was considered of low quality [9], as it did not meet two of the seven critical domains. When overlaps of the primary studies were analyzed (Figure 2), 266 primary studies appeared in at least two reviews. The degree of overlap ranged from 0% to 16% (Figure 2). One study [13] showed high values of overlap with another four studies [7,11,19,21] and moderate values with another one [12]. Another two studies showed cross-overlap, with the value reaching 14% [10,14]. Overall, the CCA showed a degree of overlap of 2.05%, which is considered low.

Regarding the risk estimates for COVID-19 hospitalization, all conditions considered in the seven meta-analyses investigating this outcome were found to be associated with increased risk (Table 2 and Figure 3a). For conditions for which we could provide a pOR, CVD showed the strongest association (pOR = 4.05; 95% CI: 3.20–5.12); the estimated pOR for CPD, CD DM, hypertension, and cancer ranged from 2.27 to 2.94, and the pOR for male sex was 1.49. Age, smoking status, obesity, and CKD were studied in <3 meta-analyses, and therefore, we did not calculate a pOR, but all the individual estimations showed an increased risk, which was higher than 2.5 for age, obesity, and CKD. The estimations of the individual meta-analyses were in a similar range for male sex, hypertension, and CVD but were more heterogeneous for cancer, DM, CPD, and CD.

For the development of severe/critical COVID-19, the highest estimated pOR was for CPD (2.82; 95% CI: 1.92–4.14); obesity, cancer, CKD, DM, and CD showed a pOR ranging from 1.91 to 2.44; CLD showed a pOR of 1.76 (95% CI 1.12–2.78); interestingly, this condition was the only one for which some individual meta-analyses could not show a significant association with the outcome. For male sex and smoking status, a pOR could not be calculated; individual studies showed a lower OR than for other epidemiological characteristics and comorbidities, in the range of 1.28–1.30 (Table 3 and Figure 3b). The estimates for each condition in the individual meta-analyses showed some heterogeneity for all of them.

Regarding mortality, three conditions showed a pOR > 3 (CKD, CVD, CD); pOR ranged from 2.24 to 2.52 for CPD, hypertension, cancer, and DM. A pOR could not be calculated for male sex, smoking status, and obesity; the OR from individual studies was in the range of 1.40 to 1.89 for these conditions (Table 4 and Figure 3c). Overall, the estimated strength of association of the different conditions with mortality was quite homogeneous across studies.

Overall, the exclusion of studies with higher degrees of overlap did not change the results. Twelve of the sixteen selected systematic reviews and meta-analyses had significant heterogeneity, and eleven systematic reviews had I2 > 50%. Individual studies in each meta-analysis differed in terms of geographic location, ethnicity of the selected subjects, frequency of diagnosis of the determinant condition, method of diagnosis, COVID-19 classification, duration of follow-up, and outcome assessment. These studies did not publish the heterogeneity of the primary studies included in the specific risk comparison.

We were unable to establish the possible publication bias according to Egger’s regression test. The test was repeated in 10 studies of meta-analyses because the remaining had insufficient data. Of the ones we reanalyzed, five systematic reviews had statistical evidence of publication bias. For the meta-analyses that could not be reanalyzed, none reported significant publication bias or did not perform or publish any statistical test of publication bias for the specific exposure comparison.

## 4. Discussion

In this umbrella review, which used restrictive criteria for the inclusion of studies, we found that male sex, age >60 years, being a smoker, and suffering from hypertension, DM, cancer, CD, CPD, CLD, CVD, and CKD are associated with significantly higher risk of hospitalization, severe disease, and death due to COVID-19. The estimations for the risks in the individual meta-analyses showed some differences but were more homogeneous for the mortality predictors.

Previous umbrella reviews also evaluated the impact of underlying conditions on COVID-19 outcomes; however, these studies had important differences with this one, the most important being that the criteria used to select the studies were less restrictive than in ours. Two of them evaluated only one underlying disease or a group of conditions. In a study on obesity, Kristensen et al. found a similar risk estimate as in our study [23]. Kastora et al. studied the impact of DM on COVID-19 outcomes [24]; the risks estimated for ICU admission and mortality in patients with DM in that study were 1.56 (95% CI: 1.28–1.89) and 1.82 (96% CI 1.65–2.02), respectively, which was somewhat lower than in our study. Harrison et al. studied the impact of cardiovascular risk factors on COVID-19 severity [25]; overall, the risk estimates in that study were similar to ours. We found one umbrella review including all the underlying conditions, focusing on whether there were geographical differences [26]. Interestingly, the authors found some regional heterogeneity in the risk estimates.

Some of the chronic conditions associated with worse outcomes in COVID-19, such as hypertension, obesity, DM, CPD, CKD, CLD, and CD, share some characteristics, including chronic proinflammatory states and innate or adaptive immunity dysfunction, which might facilitate a dysregulated immune response against SARS-CoV-2 [27,28]. Some of these conditions and smoking can also increase the expression of ACE2, the viral receptor in respiratory tract cells [29]. Age-related immune system changes (immunosenescence and inflammageing) have been associated with the increased risk of complications and mortality in older persons with COVID-19 [30]. However, from a physiopathogenic perspective, the confounding or modifying effect of age on the underlying conditions and vice versa must be considered when evaluating their independent risk estimates. As an example, male sex is also associated with severity and mortality in patients with COVID-19; although this may be related to the confounding effect of some comorbidities that might be more frequent among men, it might as well be a consequence of the sexual dimorphism in the immune response [31] and lower circulating concentration of ACE in females compared to males [32]. Regarding the smoking status, although the potential confounding effect of some associated comorbidities may play a role in the association found, smoking is known to alter mucosal innate immunity patterns and can increase the expression of ACE2 [33].

The association of cancer with deleterious outcomes in COVID-19 patients found in our study would need some considerations. Early in the pandemic, it was suggested that only patients who had recently received cancer treatment were at increased risk of death, which could be linked to treatment-related immunosuppression [34]. Additionally, patients with active hematologic malignancies seem to be at increased risk of mortality [35]. However, depending on the treatment received, some subsets of patients receiving immunomodulatory drugs, which may help avoiding the deleterious dysregulated immune response in COVID-19, might even have a better prognosis [36]. When interpreting the results of this study, it should be noted that we could only analyze the underlying conditions included in the systematic reviews detected by our strategy. Therefore, we could not evaluate the potential impact of less frequent diseases, including autoimmune conditions being treated with immunosuppressive drugs [37] or rare diseases with neuromuscular involvement [38], among others.

Our study has some limitations; because of the nature of an umbrella review, it was not possible to provide data on specific subgroups within each comorbid condition evaluated, for which the risk may be substantially different. Additionally, the heterogeneity of the systematic reviews and meta-analyses included must be considered when interpreting the data. The possible causes of that heterogeneity are differences in the designs, populations included, and methodological approaches. We could not provide the estimations for publication bias using Egger’s test, since the cumulative assessment did not exceed 10 studies. We included meta-analyses with adjusted and unadjusted estimations. We did not consider the geographical differences in the impact of the conditions. Most of the studies did not include information on in-hospital therapy nor on early treatment. Finally, most of the studies included were performed before vaccines were available and the omicron variant was predominant. However, these results could be useful in case of a rise of new virulent variants with immune-escape capacity. The strengths include studies evaluating several comorbidities and the large number of patients represented in the meta-analysis reported.

## 5. Conclusions

This umbrella review provides a comprehensive summary of meta-analyses examining the impact of patients’ characteristics on COVID-19 outcomes. Elderly patients and those with cardiovascular, cerebrovascular, and chronic renal diseases should be prioritized for pre-exposure and post-exposure prophylaxis and early treatment.

## Figures and Tables

**Figure 1 biomedicines-10-02437-f001:**
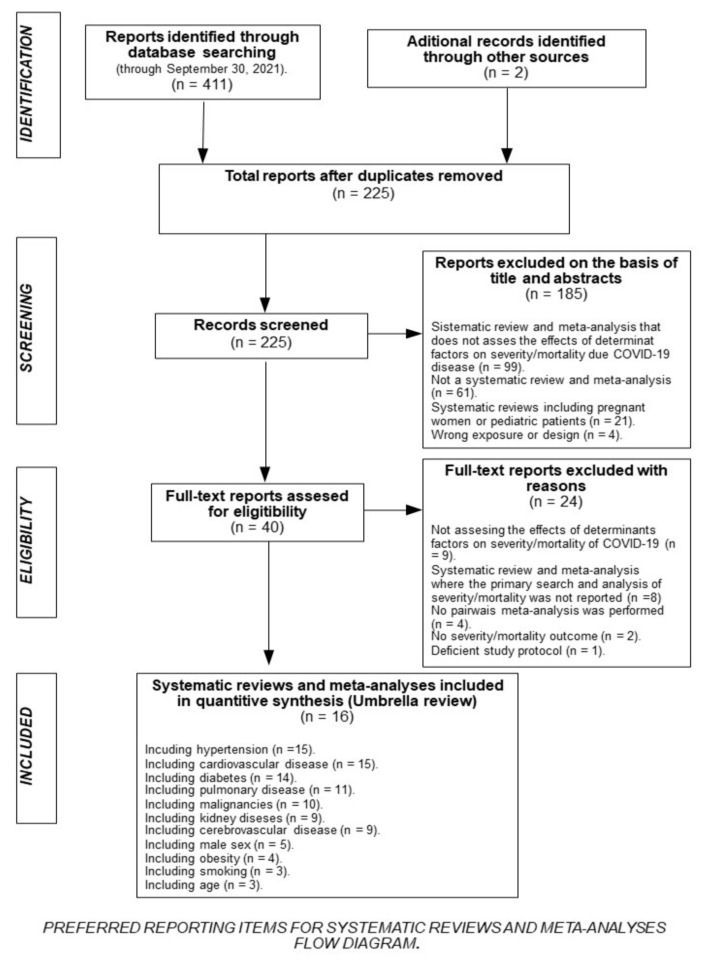
Flow chart of included articles according to PRISMA.

**Figure 2 biomedicines-10-02437-f002:**
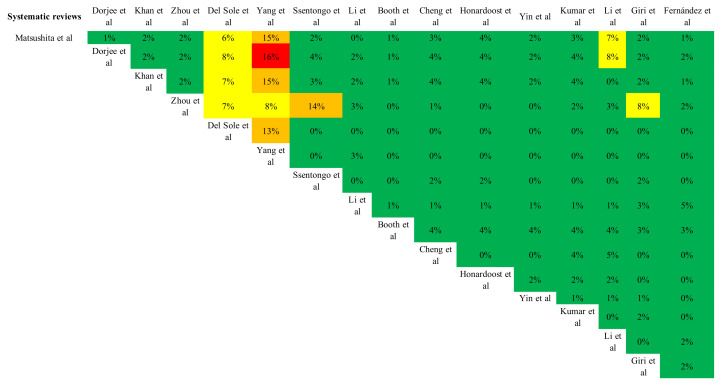
Estimation of the overlap across studies included in the umbrella systematic review quantified by using the corrected covered area. Green: light degree of overlapping (0–5%); yellow: moderate degree of overlapping (6–10%), orange: high degree of overlapping (11–15%); and red: very high degree of overlapping (>15%).

**Figure 3 biomedicines-10-02437-f003:**
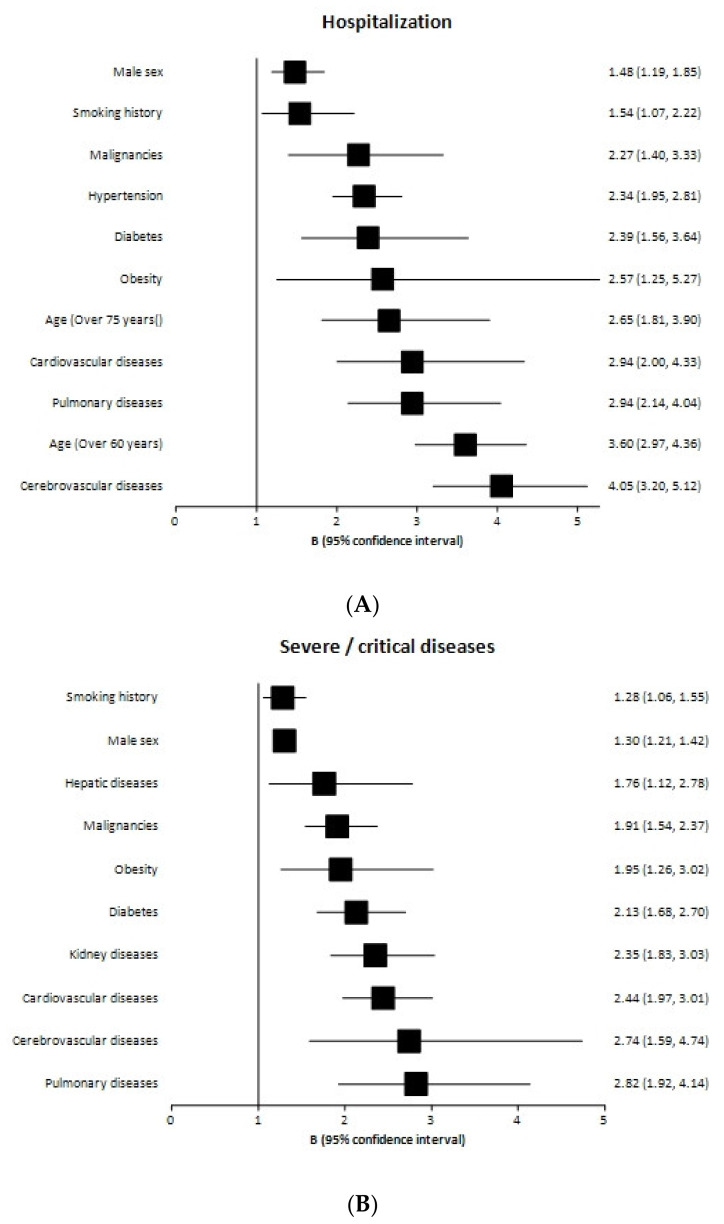
Forest plot of pooled odds ratio and 95% confidence intervals of the different patients’ characteristics and underlying conditions for (**A**) Hospitalization (**B**) Severe/critical condition, and (**C**) Mortality.

**Table 1 biomedicines-10-02437-t001:** Summary of features of the systematic reviews and meta-analyses investigating the outcome impact of patients’ characteristics and underlying conditions included in this study.

Author	Last Date of Search	Names of Databases Searched	Number of Selected Articles	Number of Selected Patients	Determinant Factors	Outcomes Related to COVID-19	Instrument of Quality Appraisal	Amstar-2 Score
Matsushita et al. [19].	3 April 2020	PubMed and Embase	25	76,638	Age, Male sex, Hypertension, DM, CD	Death	Newcastle Ottawa Quality Assessment Scale	High
Dorjee et al. [11]	31 August 2020	Medline, Embase, Web of Science, and the WHOOVID-19 database	77	38,906	Age, Male sex, Smoking, CKD, Hypertension, CLD, DM, CPD, CD	Death, severity	Newcastle Ottawa Quality Assessment Scale	High
Khan et al. [21]	1 May 2020	Medline, Web of Science, Scopus, and CINAHL	41	27,670	Malignancies, CKD, Hypertension, CLD, DM, CPD, CD, CVD	Death	Newcastle Ottawa Quality Assessment Scale	High
Zhou et al. [12].	25 April 2020	PubMed, Embase, and Cochrane Library	34	16,110	Obesity, Malignancies, CKD, Hypertension, CLD, DM, CPD, CD, CVD	Severity/Death	Not reported	High
Del Sole et al. [7]	28 May 2020	PubMed, ISI Web of Science, SCOPUS, and Cochrane databases	12	2794	Male sex, Smoking, Hypertension, DM, CPD, CD, CVD	Severity	Not reported	Moderate
Yang et al. [13]	25 February 2020	PubMed, EMBASE, and Web of Science	7	1576	Hypertension, CPD, CD	Severity	Not reported	High
Ssentongo et al. [4]	7 July 2020	MEDLINE, SCOPUS, OVID, and Cochrane Library databases and medrxiv.org	25	484	Malignancies, CKD, Hypertension, DM, CD	Mortality	Newcastle Ottawa Quality Assessment Scale	High
Li J et al. [16]	28 February 2021	PubMed, Embase, Web of Science, and Cochrane Library for epidemiological studies	41	21,060	Male sex, Smoking, Obesity, malignancies, CKD, Hypertension, CLD, DM, CPD, CD, CVD	Severity	Newcastle Ottawa Quality Assessment Scale	High
Booth et al. [22]	9 July 2020	PubMed and SCOPUS	66	1,786,001	Age, Male sex, Obesity, Malignancies	Severity	Newcastle Ottawa Quality Assessment Scale	Moderate
Cheng et al. [8]	1 April 2020	PubMed, Embase, China National Knowledge Infrastructure (CNKI), and Wanfang Database	22	3286	Malignancies, Hypertension, DM, CPD, CD, CVD	Severity	Newcastle Ottawa Quality Assessment Scale	High
Honardoost et al. [9]	28 February 2021	Electronic literature	28	6270	Hypertension, DM, CPD, CD, CVD	Severity	Newcastle Ottawa Quality Assessment Scale	Low
Yin et al. [10]	18 January 2021	PubMed, Web of Science, and CNKI	41	12,526	Malignancies, CKD, Hypertension, CLD, DM, CPD, CD, CVD	Severity	Not reported	High
Sahu et al. [15]	24 May 2020	PubMed, Embase, and Web of Science	22	4380	Obesity, Malignancies, CKD, Hypertension, DM, CPD, CD	Severity	Not reported	High
Li X et al. [20]	14 April 2020	PubMed, Embase, and Cochrane Library	12	2445	Malignancies, Hypertension, DM, CPD, CD, CVD	Severity	Newcastle Ottawa Quality Assessment Scale	High
Giri et al. [17]	20 November 2020	PubMed, Scopus, Embase, and Web of Science	41	16,495	Malignancies, Hypertension, DM, CD, CVD	Severity	Methodological Index for Non-Randomized Studies	High
Fernández et al. [18]	28 May 2020	MEDLINE, bioRXiv, and MedRXiv,	74	44,672	CKD, Hypertension, CD, CVD	Severity (One parameter for mortality)	ROBINS-I tools	High

DM: diabetes mellitus; CD: cardiac disease (including arrhythmia or chronic heart failure); CKD: chronic kidney disease; CLD: chronic liver disease; CPD: chronic pulmonary diseases; CVD: cerebrovascular diseases (including stroke and transient ischemic attack).

**Table 2 biomedicines-10-02437-t002:** Meta-analysis of the different patients’ characteristics and underlying conditions and the risk of hospitalization due to COVID-19.

Condition	Study	Number of Primary Studies	Odds Ratio	IC 95%
Male sex	De sole et al.	12	1.22	1.01–1.49
	Xinyian Li et al.	41	1.51	1.33–1.71
	Booth et al.	66	2.05	1.39–3.04
	**pOR**	**119**	**1.48**	**1.19–1.85**
Age	Dorjee et al.	77	3.60 (Age > 60 years)	2.97–4.36
	Booth et al.	66	2.65 (Age > 75 years)	1.81–3.90
Smoking history	Del Sole et al.	12	1.54	1.07–2.22
Obesity	Booth et al.	66	2.57	1.25–5.27
Malignancy	Booth et al.	66	1.46	1.04–2.04
	Cheng et al.	22	3.18	2.09–4.82
	Yin et al.	41	2.63	1.75–3.93
	**pOR**	**129**	**2.27**	**1.40–3.33**
Chronic renal disease	Yin et al.	41	3.60	2.18–5.94
Hypertension	Del sole et al.	12	2.24	1.63–308
	Cheng et al.	22	2.79	1.66–4.69
	Honardoost et al.	28	2.37	1.80–3.13
	Yin et al.	41	2.13	1.81–2.51
	**pOR**	**103**	**2.34**	**1.95–2.81**
Diabetes mellitus	Del Sole et al.	12	2.78	2.09–3.72
	Cheng et al.	22	1.64	1.30–2.08
	Honadoost et al.	28	3.18	2.09–4.82
	**pOR**	**62**	**2.39**	**1.56–3.64**
Chronic pulmonary disease	Del Sole et al.	12	2.39	1.10–5.19
	Cheng et al.	22	1.98	1.26–3.12
	Honadoost et al.	28	4.19	2.84–6.19
	Yin et al.	41	3.14	2.35–4.19
	**pOR**	**103**	**2.94**	**2.14–4.04**
Cardiovascular disease	Del Sole et al.	12	2.84	1.59–5.10
	Cheng et al.	22	1.79	1.08–2.96
	Honadoost et al.	28	4.81	3.43–6.74
	Yin et al.	41	2.76	2.18–3.49
	**pOR**	**103**	**2.94**	**2.00–4.33**
Cerebrovascular disease	Del Sole et al.	12	3.66	1.73–7.72
	Cheng et al.	22	3.92	2.45–6.28
	Honardoost et al.	28	4.85	3.11–7.57
	Yin et al.	41	3.70	2.51–5.45
	**pOR**	**103**	**4.05**	**3.20–5.12**

pOR: pooled odds ratio of included studies for each condition.

**Table 3 biomedicines-10-02437-t003:** Meta-analysis of the different patients’ characteristics and underlying conditions and the risk of development of severe/critical COVID-19.

Condition	Study	Number of Primary Studies	Odds Ratio	IC 95%
Male sex	Dorjee et al.	77	1.30	1.21–1.42
Smoking history	Dorjee et al.	77	1.28	1.06–1.55
Obesity	Zhou et al.	34	1.72	1.04–2.85
	Kumar et al.	22	2.84	1.19–6.77
	**pOR**	**56**	**1.95**	**1.26–3.02**
Malignancy	Zhou et al.	34	2.73	1.73–4.21
	Ssentongo et al.	25	1.47	1.01–2.14
	Kumar et al.	22	2.38	1.25–4.52
	Li et al.	12	2.21	1.04–4.72
	Giri et al.	41	1.75	1.40–2.18
	**pOR**	**134**	**1.91**	**1.54–2.37**
Chronic renal disease	Dorjee et al.	77	2.5	2.09–2.99
	Zhou et al.	34	3.02	2.23–4.08
	Ssentongo et al.	25	3.25	1.13–9.28
	Kumar et al.	22	1.46	1.06–2.02
	Fernadez et al.	74	2.5	1.82–3.44
	**pOR**	**232**	**2.35**	**1.83–3.03**
Chronic liver disease	Dorjee et al.	77	2.65	1.88–3.75
	Zhou et al.	34	1.54	0.95–2.49
	Yin et al.	41	1.32	0.96–1.82
	**pOR**	**115**	**1.76**	**1.12–2.78**
Diabetes mellitus	Dorjee et al.	77	1.5	1.36–1.65
	Zhou et al.	34	2.63	2.08–3.33
	Ssentongo et al.	25	1.82	1.43–2.23
	Kumar et al.	22	2.29	1.56–3.39
	Li et al.	12	3.17	2.26–4.45
	Giri et al.	41	2.04	1.67–2.50
	**pOR**	**211**	**2.13**	**1.68–2.70**
Chronic pulmonary disease	Dorjee et al.	77	1.7	1.4–2.0
	Zhou et al.	34	3.56	2.87–4.41
	Yang et al.	7	2.46	1.76–3.44
	Kumar et al.	22	2.92	1.70–5.02
	Li et al.	12	5.08	2.68–9.63
	**pOR**	**152**	**2.82**	**1.92–4.14**
Cardiovascular disease	Dorjee et al.	77	2.1	1.82–2.43
	Zhou et al.	34	3.13	2.65–3.70
	Ssentongo et al.	25	2.25	1.60–3.17
	Kumar et al.	22	1.61	1.31–1.98
	Li et al.	12	2.66	1.71–4.15
	Giri et al.	41	2.78	2.00–3.86
	Fernández et al.	34	3.20	2.29–4.48
	**pOR**	**245**	**2.44**	**1.97–3.01**
Cerebrovascular disease	Zhou et al.	34	2.74	1.59–4.74

pOR: pooled odds ratio of included studies for each condition.

**Table 4 biomedicines-10-02437-t004:** Meta-analysis of the different patients’ characteristics and underlying conditions and the risk of mortality due to COVID-19.

Condition	Study	Number of Primary Studies	Odds Ratio	IC 95%
Male sex	Matsushita et al.	25	1.73	1.50–2.01
Smoking history	Xinyang et al.	41	1.40	1.06–1.85
Obesity	Xinyang et al.	41	1.89	1.44–2.46
Malignancy	Kahn et al.	41	2.22	1.63–3.03
	Xinyang et al.	41	2.60	2.00–3.40
	**pOR**	**82**	**2.43**	**1.99–2.97**
Chronic kidney disease	Khan et al.	41	3.02	2.60–3.51
	Li et al.	41	2.97	1.63–5.41
	**pOR**	**82**	**3.02**	**2.61–3.49**
Hypertension	Matsushita et al.	25	2.87	2.09–3.93
	Li et al.	41	2.42	2.03–2.88
	**pOR**	**66**	**2.52**	**2.16–2.94**
Chronic liver disease	Kahn et al.	41	2.35	1.50–3.6
	Li et al.	41	1.51	1.06–2.17
	**pOR**	**82**	**1.85**	**1.20–2.85**
Diabetes mellitus	Matsushita et al.	25	3.20	2.26–4.53
	Kahn et al.	41	2.46	2.03–2.85
	Li et al.	41	2.40	1.98–2.91
	**pOR**	**107**	**2.52**	**2.22–2.85**
Chronic pulmonary disease	Khan et al.	41	1.94	1.72–2.19
	Li et al.	41	2.88	1.89–4.38
	**pOR**	**82**	**2.24**	**1.54–3.25**
Cardiovascular disease	Matsushita et al.	25	4.97	2.76–6.58
	Ali Khan et al.	41	3.42	2.86–4.09
	Yang et al.	7	3.41	1.88–6.22
	Li et al.	41	2.87	2.22–3.71
	**pOR**	**114**	**3.59**	**2.83–4.56**
Cerebrovascular disease	Khan et al.	41	4.12	3.04–5.58
	Li et al.	41	2.47	1.54–3.97
	Giri et al.	41	2.68	1.29–5.57
	Fernández et al.	75	2.70	1.74–4.19
	**pOR**	**198**	**3.11**	**2.36–4.11**

pOR: pooled odds ratio of included studies for each condition.

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
