# Peer review of "Association of Patients’ Epidemiological Characteristics and Comorbidities with Severity and Related Mortality Risk of SARS-CoV-2 Infection: Results of an Umbrella Systematic Review and Meta-Analysis"

_biomedicines, 2022, doi:10.3390/biomedicines10102437_

Round 1

Reviewer 1 Report

This article addresses a very important topic on the importance of pre-existing comorbidities in patients COVID-19. The authors state that some comorbidities carried a worse outcome after Sars-CoV2 infection. I think that this article is very interesting and useful; the design of the study is good. The results are in line with the discussion and methods. I only have some comments:

-Introduction/background. What about patients affected by autoimmune disease who are treated with immunosuppressants? (i.e., Myasthenia gravis, IBD, transplant receivers, etc.). This topic is fully discussed in a very recent study exploring the impact of COVID-19 in fragile patients affected by Myasthenia Gravis (https://doi.org/10.3390/ neurolint14020033). These results underlined the fact that MG patients presented high mortality from COVID-19, especially unvaccinated patients. I suggest to report these recent data and add them to the introduction briefly and also discuss their importance on the light of the role of chronic pulmonary disease and respiratory insufficiency in the prognosis of COVID-19. 

-Results/Discussion. Data on rare diseases are missing. There is a significant risk of underestimation of some conditions in which COVID-19 infection might represent a severe cause of deterioration and, in some cases, the cause of death. This is the case of severely ill patients, patients with severe immune-deficits, and patients with respiratory insufficiency (i.e. myasthenia gravis. myotonic dystrophy, Pompe disease, ALS, and many neuromuscular conditions). Indeed, patients affected by neuromuscular disease are often not considered in clinical routine, but they represent a challenge for the clinician. We all can remember that these special populations have been considered with priority by vaccination programs (i.e., Italian campaign). Moreover, recent literature has considered the impact of pandemic in these special populations (https://pubmed.ncbi.nlm.nih.gov/32661716/), but data on COVID-19 and vaccines are still missing apart from single reports and case series. Hence, the manuscript should have a specific section (“Special considerations”) considering the role of neuromuscular comorbidity (in terms of respiratory insufficiency, mechanic ventilation, ICU admission) in COVID-19. 

-Limitations. The assessment of bias is not complete. This strengthens previous suggestions. Add a sentence regarding the risk of underestimation the role of comorbidities in special populations.

-Grammar and style are adequate.

Author Response

To the Editorial office of

Biomedicines

Seville September 7, 2022

Dear Prof. Olivera Petrovic

We would like to thank you for the opportunity to resubmit a revised version of our manuscript biomedicines-1901392 entitled “Association of patients’ epidemiological characteristics and comorbidities with severity and related mortality risk of SARS-CoV-2 infection: results of an umbrella systematic review and meta-analysis”. We have carefully considered every comment and suggestion raised by the reviewers and changed our manuscript accordingly. Our point-by-point responses are provided below. We attach two copies of the revised manuscript, one of them with all the changes clearly marked in yellow.

Q= QUERY; A= ANSWER

Reviewer: 1

Comments to the Author

Q1. English language and style are fine/minor spell check required.

A1. We have thoroughly revised the text and corrected some grammatical defects and spelling mistakes.

Q2. Does the introduction provide sufficient background and include all relevant references? it must be improved.

A2. We included some changes in the introduction.

Q3. Introduction/background. What about patients affected by autoimmune disease who are treated with immunosuppressants? (i.e., Myasthenia gravis, IBD, transplant receivers, etc.). This topic is fully discussed in a very recent study exploring the impact of COVID-19 in fragile patients affected by Myasthenia Gravis (https://doi.org/10.3390/ neurolint14020033). These results underlined the fact that MG patients presented high mortality from COVID-19, especially unvaccinated patients. I suggest to report these recent data and add them to the introduction briefly and also discuss their importance on the light of the role of chronic pulmonary disease and respiratory insufficiency in the prognosis of COVID-19.

A3. We appreciate the reviewer's comment regarding underlying diseases in which some type of pharmacological immunosuppression is involved (MG, IBD or transplant receivers). Because we performed an umbrella systematic review, we could only include the underlying conditions included on the systematic review detected with our literature search strategy; the conditions specified by the reviewer were not considered in those. However, we agree with the reviewer about the importance of having these conditions and others into account, and therefore we include a comment in the Discussion section and two references, including this as a limitation (lines 236-240).

Q4. Results/Discussion. Data on rare diseases are missing. There is a significant risk of underestimation of some conditions in which COVID-19 infection might represent a severe cause of deterioration and, in some cases, the cause of death. This is the case of severely ill patients, patients with severe immune-deficits, and patients with respiratory insufficiency (i.e. myasthenia gravis. myotonic dystrophy, Pompe disease, ALS, and many neuromuscular conditions). Indeed, patients affected by neuromuscular disease are often not considered in clinical routine, but they represent a challenge for the clinician. We all can remember that these special populations have been considered with priority by vaccination programs (i.e., Italian campaign). Moreover, recent literature has considered the impact of pandemic in these special populations (https://pubmed.ncbi.nlm.nih.gov/32661716/), but data on COVID-19 and vaccines are still missing apart from single reports and case series. Hence, the manuscript should have a specific section (“Special considerations”) considering the role of neuromuscular comorbidity (in terms of respiratory insufficiency, mechanic ventilation, ICU admission) in COVID-19. 

A4. Please see our answer to Q3.

Q5. Limitations. The assessment of bias is not complete. This strengthens previous suggestions. Add a sentence regarding the risk of underestimation the role of comorbidities in special populations. 

A5. Please see our response to previous comments.

Reviewer 2 Report

Thank you for giving me the opportunity to read and comment a report “Association of patients’ epidemiological characteristics and comorbidities with severity and related mortality risk of SARS-2 CoV-2 infection: results of an umbrella systematic review and meta-analysis.”, by Reyna-Villasmil E, et al.

In the reviewed manuscript, the association between patients´ epidemiological characteristics and comorbidities with severity and related mortality risk of SARS-CoV-2 infection has been investigated. 

This paper is well written, correctly structured with a suitable research concept, the study limitations are addressed, and it is of relevance to readers of the journal. However, I include a comments for your consideration.

·       According to the rules of the journal the abstract must be without headings.

·       Abbreviations should be defined in all tables of the manuscript (footnotes).

·       The quality of figure 2 should be improved.

·       The abbreviation ODD appears in Table 2 and Figure 3. What is its meaning? If it is an error, it should be corrected.

·       In table 2 and table 4, the IC should be in a single line, as in table 3.

·       The titles of table 2, 3 and 4 should be more explanatory.

·       Figure 3 is not complete. Furthermore, the quality should be improved.

Author Response

Reviewer: 2

Q6. English language and style are fine/minor spell check required.

A6. We have thoroughly revised the text and corrected some grammatical defects and spelling mistakes.

Q7.  According to the rules of the journal the abstract must be without headings

A7. We revised the abstract and deleted the headings in the new version as requested. This answer is included in lines 30-49 in the full text.

Q8. Abbreviations should be defined in all tables of the manuscript (footnotes). 

A8. All abbreviations have been reviewed and corrected.

Q9. The quality of figure 2 should be improved.

A9. Quality of figure 2 has been improved.

Q10. The abbreviation ODD appears in Table 2 and Figure 3. What is its meaning? If it is an error, it should be corrected.

A10. Thank you. This error has been corrected (underlined in yellow in both figures).

Q11.  In table 2 and table 4, the IC should be in a single line, as in table 3.

A11. We revised the tables in our file, the IC appears in a single line in the 3 tables.

Q12. The titles of table 2, 3 and 4 should be more explanatory

A12. Thank you for emphasizing this point. The titles of the tables have been corrected and highlighted at the top of each table.

Q13. Figure 3 is not complete. Furthermore, the quality should be improved.

A13. Thank you for your comment. We revised the Figure and we cannot find the aspect that is missing. We would be happy to revise again if this is specified. We have also improved the quality of the graphics.